# omg2letters: Translating Muscle Activity into Written Language

Muhammad Salman Kabir
*Vladimir Zelman CNBR*
*Skoltech*
Moscow, Russia
ORCID: 0000-0002-0071-6901

Mikhail Lebedev
*Faculty of Mechanics & Mathematics*
*Lomonosov Moscow State University*
Moscow, Russia
ORCID: 0000-0003-0355-8723

Gurgen Soghoyan
*Vladimir Zelman CNBR*
*Skoltech*
Moscow, Russia
ORCID: 0000-0002-0034-0326

*Abstract*—Despite significant advancements in brain-computer interface (BCI) technology, systems capable of leveraging physiological signals to detect and recognize human intentions in real-time are still underdeveloped. To achieve a new level of human-machine interaction, it is essential to integrate motor activity correlates with state-of-the-art artificial intelligence (AI) architectures. In this study, we present the first demonstration of handwriting decoding − a complex motor task − using a novel myographic method called Optomyography (OMG). Unlike previous electromyography (EMG)-based approaches that treat handwriting decoding as a classification problem, we frame it as a continuous trajectory reconstruction challenge. We evaluated GRUScribe (GRU-based decoder) and TransScribe (transformer-based decoder), successfully decoding 10 numerical digits and 33 Russian letters from 20 able-bodied and 4 amputee participants, without requiring elaborate preprocessing. Our results demonstrate the remarkable potential of OMG for recognizing complex motor activity. We believe that our work sets a new benchmark in non-invasive muscle activity decoding, offering direct applications in advanced prosthetic control and human-machine interfaces.

*Index Terms*—Optomyography, Machine learning, Augmentation, Motor decoding, Electromyography, Human-machine interaction, Brain computer interfaces

## I. INTRODUCTION

Daniel Wolpert's profound observation cuts to the core of our existence: "We have a brain for one reason and one reason only and that is to produce adaptable and complex movements. Movement is the only way you have of affecting the world around you." [1]. This elegant statement captures a fundamental truth about human existence − our ability to interact with and shape our environment relies entirely on movement. If movement is our primary means of engaging with the world, then limb amputation represents a profound disruption to our most basic way of being human.

Globally, limb amputation represents a major health crises as it affects millions of people, yearly [2]. Loss of limb functionality greatly impacts personal independence, professional capabilities and psychological health [2], [3]. Prosthetic technologies emerge as critical interventions to restore motor function, enable daily activities and mitigate the physiological and psychological consequences of limb loss. Historically, prosthetic development has progressed through multiple technological stages to restore limb functionality [4]. Early mechanical prostheses primarily served cosmetic purposes while providing basic structural support with minimal movement capabilities. Later, body-powered prosthetic systems introduced functional movement by manipulating residual limb muscle activity to generate mechanical actions. However, these approaches offered limited degrees of freedom and thus limited control [4].

Electromyography (EMG) technology caused a paradigm shift in prosthetic design [5]–[7]. This technology enables interpretation of neuromuscular signals by capturing electrical potentials generated during muscle contractions [8]. By translating bioelectrical signals into mechanical commands, EMG-based prostheses offer remarkable potential for more natural and responsive artificial limb control. Nevertheless, current EMG-based prosthetic systems face substantial challenges in terms of limited signal-to-noise ratio, cross talk among channels and insufficient sensory feedback [5]–[7], [9]–[11]. Moreover, patients frequently report difficulties in precise device control, mental fatigue during operation and a persistent gap between prosthetic performance and natural limb functionality [5]–[7]. These limitations constrain the functional utility and user acceptance of advanced prosthetic technologies.

Apart from prosthetic control system, myographic signals have substantially contributed to the evolution of human-machine interfaces (HMIs) [12], [13]. These systems enable users to interact with or/and control a diverse array of external technologies. A growing trend in this domain is the integration of wearable devices, such as wristbands equipped with myography sensors, that interpret muscle activity to issue control commands [14]. However despite notable advancements in hardware design, signal fidelity and machine learning-based physiological signal decoding techniques, myographic-based HMIs have yet to achieve widespread adoption because of the shortcoming discussed above [12]–[14].

Recently, optomyography (OMG) [15], [16], emerged as a highly efficient non-invasive method to record muscle activity. This technique uses near-infra-red light to measure perfusion and oxygenation in a contracting muscle. Biological tissues demonstrate a unique light transmission characteristic within the 700-900 nm wavelength spectrum [17]. In a standard measurement configuration, an LED generates near-infrared

Supplementary materials: https://github.com/5a7man/omg2letters

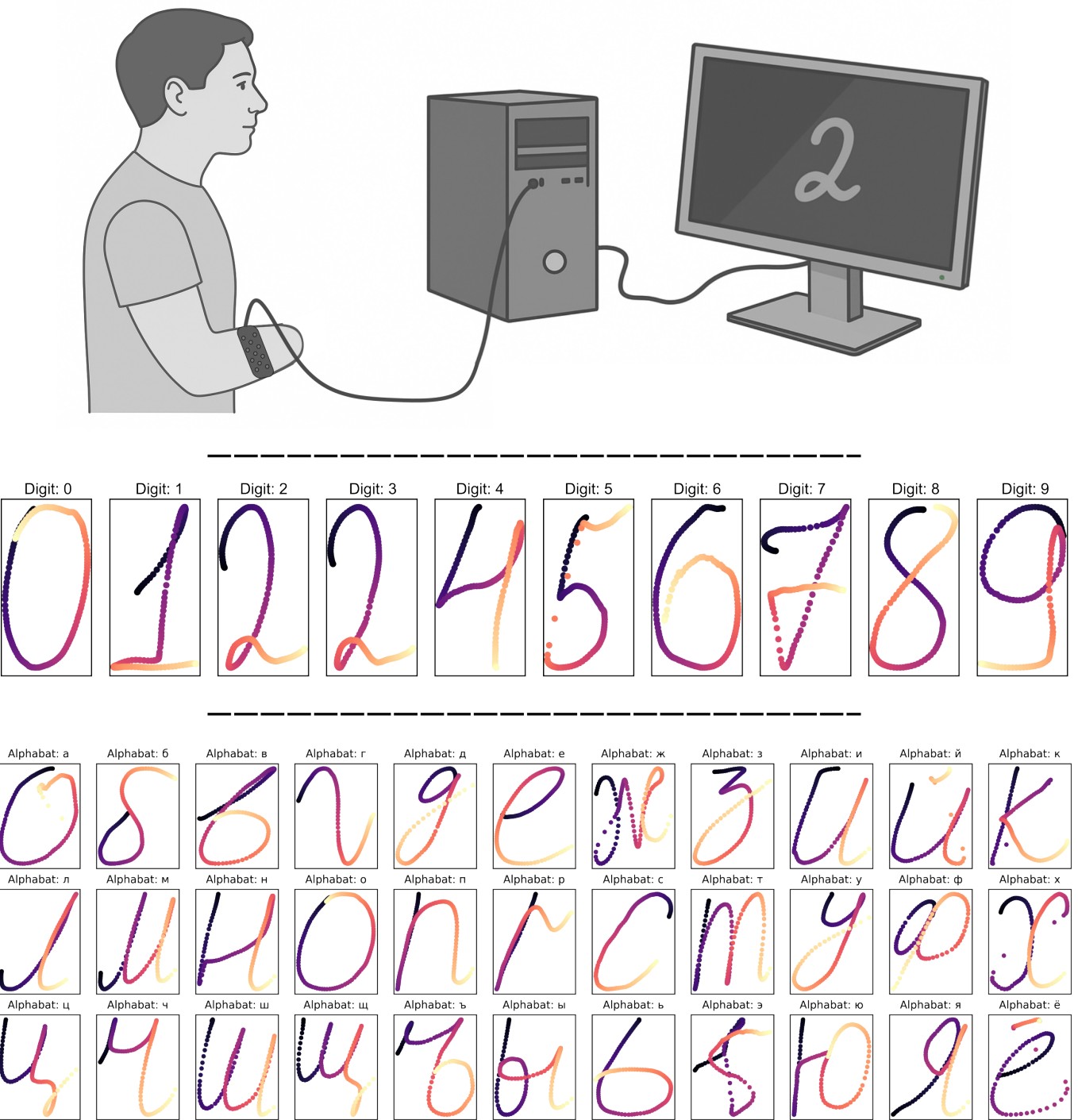

Fig. 1: **Top.** Experimental paradigm; a trans-radial amputee is replicating the trajectory of displayed digit. OMG band is on the residue limb such that the sensors are on the ventral surface of the skin. **Middle.** Decoded digits of an amputee using TransScribe. Although "3" is decoded incorrectly but overall performance is reliable. **Bottom.** Decoded alphabets of an amputee using TransScribe with reliable performance.

that penetrates the tissue, while a photodetector captures the resulting light scattering within the surrounding biological structures [16]. OMG offers several significant advantages over commonly used surface electromyography (sEMG), including

superior signal to noise ratio and higher resistance to external interference [16].

Several studies have investigated OMG for motor decoding tasks [13], [18]–[23]. However, these studies have largely

been limited to recognition of simple hand movement or discrete gestures. In this work, we advanced significantly by demonstrating OMG's potential in recognizing complex motor activity through decoding handwriting, a task requiring fine motor control and temporal precision.

Our work makes the following key contributions:

- We present the first application of decoding 10 numerical digits and 33 Russian alphabets using OMG and demonstrates OMG's untapped potential beyond simple gesture recognition.
- We evaluate state of the art sequence modeling architectures (GRU and transformer) and establish that our approach may work effectively for both **able-bodied** and **trans-radial amputees** which represents a significant advancement for assistive technology applications.
- We test data augmentation techniques tailored to OMG in our attempt to substantially improve decoding performance.

Our work establishes a new benchmark in non-invasive muscle activity decoding, with direct applications to advanced prosthetic control systems and HMIs. The capabilities demonstrated in this work, particularly the successful decoding of complex motor patterns in participants with amputated limb, represent a significant step toward bridging the gap between artificial and natural limb functionality.

## II. METHODOLOGY

This section present a comprehensive framework for decoding handwritten trajectories from OMG. Our cohort consisted of 24 participants, including 20 able-bodied individuals and 4 trans-radial amputees. All participants were over 18 years of age and provided written informed consent in accordance with protocols approved by the local ethical committee.

### A. OMG to digits

*1) Experimental paradigm:*
Our designed experimental paradigm involved displaying a five-second video of a numerical digit being drawn steadily on a computer screen. Participants were instructed to replicate the trajectory of character as shown in the presented video. Able-bodied participants used a pen to physically replicate character trajectories on a tablet or paper, while amputees exerted mental effort to move their phantom limb in a way that matched the

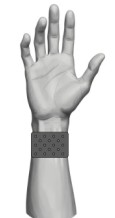

Fig. 2: An illustration of OMG band's placement on able-bodied participants

trajectory displayed on the screen. During these attempted movements, visible contractions of degenerated muscles were observed in the residual limb. Digits from 0 to 9 were displayed in a random order. Fig. 1 illustrates the experimental setup and procedure. 12 participants, including 10 able-bodied individuals and 2 trans-radial amputees participated in this experimental paradigm. This deliberate inclusion of amputees enables us to evaluate the clinical translatability of our approach.

OMG data was captured using a custom-designed OMG wristband (Motorica LLC), which incorporated four IR emitters and ten IR receivers. The wristband was capable of transmitting data across 50 channels at a frequency of 30 Hz. Figure 1 illustrates the position of wristband on amputees. For able-bodied participants, we positioned wristband on the distal forearm near carpal canal such that the sensors were on the ventral surface of the skin (Fig. 2). Each participant performed 20 trials per digit.

Experiments were conducted under direct supervision of a researcher who visually confirmed trajectory adherence in real-time. Additionally, participants were instructed to verbally report any instances where they failed to accurately track a character trajectory. This was done to ensure data quality and prevent artifacts from unintended movements.

*2) Preprocessing:* A key innovation in our approach is the intentional minimization of preprocessing steps. This design choice serves two purposes: (i) to test the inherent information carrying capacity of raw OMG, and (ii) to demonstrate the superiority of OMG over traditional EMG that typically requires extensive preprocessing. We implemented only essential channel quality assessment and discard channels that failed to meet the following empirically derived criteria:

- $\mathbb{E}[X_i] > 200$ (threshold empirically derived from pilot data) and,
- $2(\mathbb{E}[X_{i_1}]) > \mathbb{E}[X_{i_2}]$ and,
- $2(\mathbb{E}[X_{i_2}]) > \mathbb{E}[X_{i_1}]$

Here $\mathbb{E}[X_i]$ is the mean of channel $X_i$ where $i \in [1, 50]$. Similarly, $\mathbb{E}[X_{i_1}]$ and $\mathbb{E}[X_{i_2}]$ are the mean of first half and second half of channel $X_i$, respectively. This filtration approach was implemented to address potential band displacement during data collection, which could compromise signal integrity. Inadequate band-to-skin contact allows external light infiltration, which results in increased signal amplitude and reduced data quality. Apart from this minimized filtration, no experimental calibration was required or done.

*3) Augmentation:* We implemented an offline augmentation pipeline in our attempt to increase the diversity of OMG data, enhance decoders' generalization and improve the quality of decoded trajectories. This pipeline included addition of gaussian noise and fourier transform (FT) surrogates, and performing smooth time masking, sign flipping, frequency shifting, channel shuffling and time reversing in/on training data. These augmentation techniques are inspired from parallel EEG and audio processing studies. A short description of all the augmentation techniques used in this study is provided in supplementary material section 1.1.

*4) Decoder architecture:* Since our work represents the first attempt to decode handwriting using OMG, we implemented

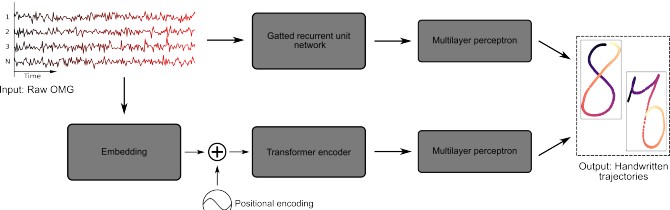

Fig. 3: **Decoders' architecture.** Raw OMG after preprocessing was fed to GRUScribe to translate into digit trajectories. Similarly, digits and alphabets trajectories were obtained by first passing OMG to embedding layer followed by TransScribe.

two state-of-the-art sequence modeling architectures to establish strong baselines for future research. Both architectures treat the decoding task as a regression problem and mapped standardized OMG data directly to continuous 2D trajectories. While these architectures are standard, their application to OMG decoding is novel.

We used Gated Recurrent Unit (GRU) [24] and Transformer [25] neural networks followed by multilayer perceptron (named as **GRUScribe** and **TransScribe**, respectively) to translate OMG data into the corresponding trajectories of digits. GRU is a type of recurrent neural network (RNN) [26] that is particularly well-suited for sequence prediction tasks due to its ability to capture temporal dependencies in time-series data. GRUs utilize gating mechanisms to control the flow of information. This control allows them to retain relevant information over longer sequences while mitigating the vanishing gradient problem, we commonly encounter in traditional RNNs.

On the other hand, the Transformer architecture, which has gained prominence in natural language processing domains, leverages self-attention mechanisms to process input data in parallel rather than sequentially. This allows for more efficient training and the ability to capture long-range temporal dependencies within the data. Fig. 3 presents the architectures of the both decoders, used in our study. Details about GRUScribe and TransScribe parameters are provided in supplementary material section 1.3.

In our implementation, the input to the decoders consisted of the standardized OMG data (having a mean of 0 and standard deviation of 1), while the output was the respective trajectories of the digits. We utilized Soft Dynamic Time Warping (SoftDTW) [27] as the loss function, while ADAMW [28] optimizer was employed to enhance the training process by adjusting the learning rate dynamically. The decoders were trained in each subject independently assuming individual set of weights for each participant. The training lasted for a predefined number of epochs (ranging from 200 to 400) without using techniques such as reducing learning rate or early stopping. We encapsulated the whole training and testing process in Leave-One-Out-Cross-Validation (LOOCV) [29] to ensure robust evaluation of the model's performance.

In LOOCV, for each participant, during each iteration, one trial was set aside for testing while the model was trained on the remaining 19 trials. This process was repeated 20 times so that each trial was used once as the test set (see supplementary material section 1.2).

*5) Evaluation metric:* To quantitatively assess the quality of the decoded/reconstructed trajectories, we utilized the Normalized Fréchet Distance (NFD) [30] as our evaluation metric. The NFD is a scale-invariant measure of similarity between two curves, derived from the standard Fréchet distance. It quantifies the minimum 'continuous' distance required to traverse both curves while considering their shapes and the order of points along the trajectories. This metric is particularly useful in our study (compared to traditional Mean Square Error (MSE) and correlation coefficients) since it accounts for variations in speed, size and timing. This allows for a more accurate comparison of the decoded/reconstructed and actual trajectories. Using test set as ground truth, we computed NFD between the decoded trajectories and ground truth for each participant.

*6) Comparative analysis:* In the final stage, we conducted a comparative analysis to evaluate the quality of trajectories decoded by GRUScribe and TransScribe. This analysis aimed to identify the most effective decoder for our task of decoding handwriting. Additionally, we performed a similar analysis to assess the quality of trajectories obtained through various augmentation methods applied to each decoder. This analysis was done to determine which augmentation techniques yielded the best and worst results in enhancing the decoder's performance.

### B. OMG to alphabets

We developed a similar experimental paradigm as in section II-A1 but instead of digits, 33 letters of Russian Cyrillic alphabet were displayed in random order. We collected OMG data from 12 participants, comprising 10 able-bodied individuals and 2 trans-radial amputees with the same custom-designed OMG wristband. It is noteworthy that although both amputees were right-hand dominant, the experimental trials were conducted using their left residual limb. Each participant performed 20 trials per letter.

For translating OMG data to corresponding alphabet trajectories, we utilized TransScribe (section. II-A4, fig. 3). Whole training and testing pipeline was encapsulated into 5-fold cross validation scheme [31]. In this validation strategy, for each participant, the 20 trials were randomly divided into five folds, each containing 4 trials. In each iteration, one fold (4 trials) was used for testing while the model was trained on the remaining 16 trials. This process was repeated five times so that each trial was used once as part of the test set (see supplementary material section 1.2). Other experimental settings remained consistent as of section II-A4. We applied the same augmentation pipeline (section II-A3) and employed an identical evaluation and comparison strategy as in sections II-A5 and II-A6, respectively.

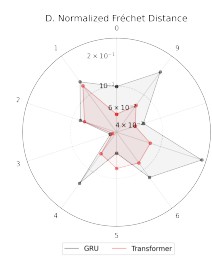

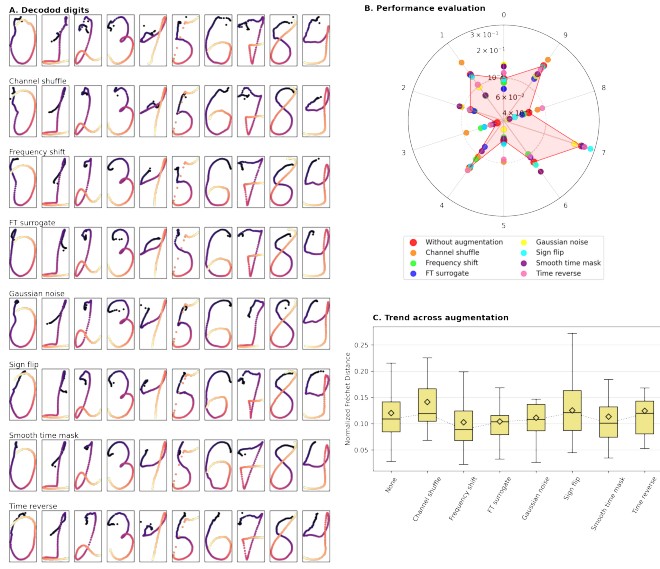

Fig. 4: Decoding results of an amputee. **A.** Ground truth used to train decoders, **B.** Decoded trajectories using GRUScribe, **B.** Decoded trajectories using TransScribe. A clear improvement in decoding can be observed, **D.** Mean NFD (over all folds) between ground truth and decoded trajectories on a log scale. The axis represent digits. Quantitatively, the TransScribe outperformed GRUScribe.

*C. A step toward real-time decoding*

In our attempt toward real-time decoding, we developed a sequential two-model approach where the first model (a transformer encoder) was trained to decode a small OMG segments into an initial trajectory of character and the second model (an RNN) was designed to forecast the subsequent trajectory based on the first model's output. We fed non overlapping segment of varying OMG sizes: initial 20%, 33%, and 50% of the total trial duration.

## III. RESULTS

We achieved significant decoding accuracy using GRUScribe with ADAMW optimizer and SoftDTW loss function. Figure 4A and 4B present the actual and decoded trajectories for an amputee, respectively. Decoded trajectories for all subjects are available in supplementary material, section 1.4. TransScribe yielded notable improvements in decoding accuracy. Figure 4C presents the decoded trajectories for an amputee with TranScribe, demonstrating enhanced trajectory decoding compared to GRUScribe. Our results with decoding of amputees was particularly encouraging. Decoded trajectories for all subjects are available in supplementary material section 1.5.

Direct comparison between GRUScribe and TransScribe revealed consistent performance differences (fig. 4B and 4C). NFD metric quantitatively validated the superior performance of TransScribe across all participants. Figure 4D present the average NFD across all cross-validation folds for each digit for an amputee. Complete comparisons for all subjects are available in supplementary material section 1.6.

Data augmentation techniques applied to GRUScribe produced variable results (fig 5A). Figure 5B presents performance metrics with and without augmentation across each digit for an amputated participant. Complete comparisons for all subjects are available in supplementary material section 1.3. Analysis revealed inconsistent patterns, with certain augmentation techniques improving performance for specific

Fig. 5: Effect of augmentation on GRUScribe performance for an amputee. **A.** Visualization of decoded trajectories with different augmentation techniques, **B.** Mean NFD between ground truth and decoded trajectories on a log scale. The axis represent digits, **C.** Box plot presenting the summary of mean NFD (over all digits) across all folds. Diamonds represent "mean" statistic. Decoder performed inconsistently for every augmentation technique.

digits while decreasing performance for others (fig 5C). Visual inspection suggested improved decoding quality except when using time reversal augmentation (see supplementary material section 1.4). We observed a similar variability when applying augmentation techniques to TransScribe (fig 6). Channel shuffle augmentation notably decreased performance of TransScribe, while other augmentation techniques showed digit-specific effects (see supplementary material section 1.5).

Extension to alphabet decoding task revealed similar trends. Figure 1 displays the decoding performance across all 33 Russian alphabets for an amputated participant. Complete trajectory comparisons for all subjects are available in supplementary material section 1.7 . While NFD showed inconsistent patterns (fig. 7A and 7B), visual inspection of trajectories indicated performance improvement (see supplementary material section 1.7).

The inconsistent effects of augmentation across decoders, digits and alphabets suggest complex interactions between OMG data and model architecture. These variations likely stem from individual physiological differences and motor-behavior variability and indicate a need for customized augmentation strategies.

The results we obtained in our attempt toward real time decoding highlighted the inter-model dependency where the second model's performance demonstrated a strong correlation with the first model's decoding accuracy (fig. 8). While the 33% and 50% window configurations forecasted reliable tra-

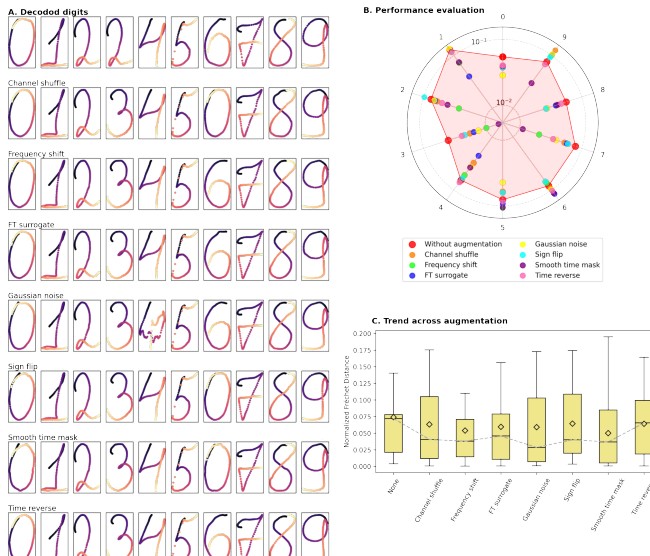

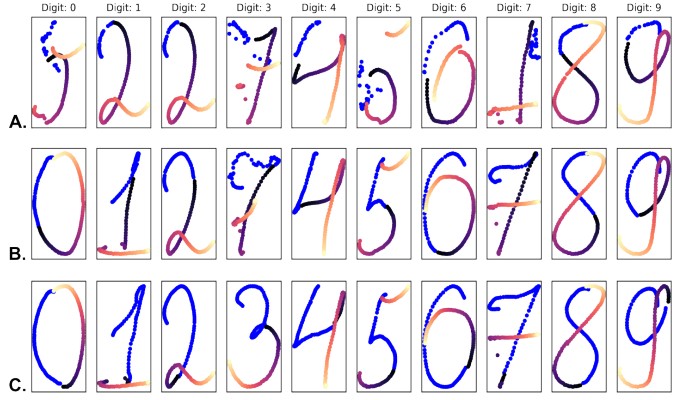

Fig. 6: Effect of augmentation on TransScribe performance for an amputee. **A.** Visualization of decoded trajectories with different augmentation techniques, **B.** Mean NFD (over all folds) between ground truth and decoded trajectories on a log scale. The axis represent digits, **C.** Box plot presenting the summary of mean NFD (over all digits) across all folds. Diamonds represent "mean" statistic. Inconsistent performance can be observed.

Fig. 8: A step toward real-time decoding. **A.** Trajectory forecasting using only the initial 20% of OMG signal, **B.** initial 33%, **C.** initial 50%. Blue represents initial trajectory.

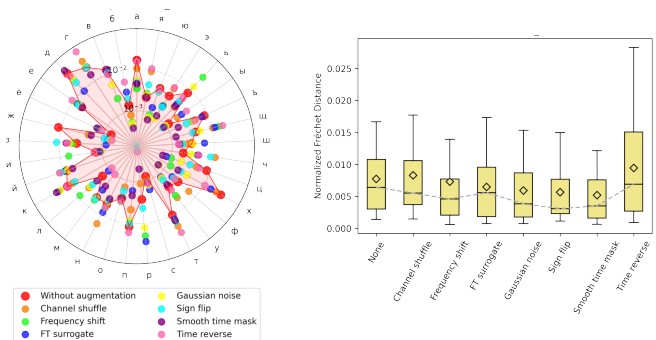

Fig. 7: Effect of augmentation on alphabets decoding of an amputee. **Left.** Mean NFD (over 5 folds) between ground truth and decoded trajectories on a log scale. The axis represent alphabets, **Right.** Box plot presenting the summary of mean NFD (over all alphabets) across 5 folds. Diamond represent "mean" statistic. Similar to digit decoding, decoder performed inconsistently here.

jectories, the 20% window consistently underperformed. This finding indicates the critical importance of refining the first model.

## IV. DISCUSSION

This study presents the first demonstration of complex handwritten trajectory decoding from OMG and established a new benchmark in non-invasive muscle activity decoding.

Our findings reveal that OMG holds significant capacity to recognize intricate muscle activity which can be decoded without elaborate preprocessing or subject-specific algorithmic customization.

We choose Russian Cyrillic alphabets alongside numerical digits; (i) to demonstrate that OMG can capture intricate muscle movements required for complex motor actions and (ii) because all participants were native to the Russian writing system. The fundamental principle that OMG can decode fine motor intentions should generalize across writing systems. However, we acknowledge that different scripts may present unique challenges, and cross-linguistic validation represents an important direction for future research

Our choice of GRU-based (GRUScribe) and Transformer-based (TransScribe) architecture was motivated by the need to capture short-long range temporal dependencies in handwriting trajectories. The superior performance of TransScribe over GRUScribe can be attributed to two architectural advantages. First, transformers' self-attention mechanism directly model relationships between any positions in the sequence regardless of distance and thus captures the long-range dependencies essential in handwriting where the beginning of a character often relates to its ending. Secondly, the multi-headed attention allows simultaneous focus on different aspects of muscles movement and thus better handles the variable timing patterns inherent in natural handwriting. These advantages make TransScribe particularly well-suited for decoding the complex temporal relationships embedded in OMG. Apart from GRUScribe and TransScribe, we explored a broader range of architectures including CNN, RNN and LSTM models, as well as hybrid Transformer+CNN architectures. However, GRUScribe and TransScribe yielded the most significant results and are therefore featured in this work.

We observed inconsistent effects of augmentation techniques across different decoders, digits and alphabets. This inconsistency suggests complex interactions between OMG data and decoders architecture. The observed variability in decoding performance across augmentation techniques may

be attributed to multiple factors. Individual differences in OMG band contact, tissue composition and participant-specific muscle activation patterns likely influence augmentation effectiveness. Furthermore, the complexity of movement execution, potential learning effects during experimental trials and muscle fatigue can contribute to the inconsistent augmentation outcomes. These factors collectively suggest that optimal augmentation strategies may need customization based on both architecture type and individual physiological characteristics. Despite this variability, our results demonstrate significant progress in using OMG to decode complex and intricate muscle movements without elaborate and complex preprocessing pipelines.

Prior work in myographic-based handwriting recognition has predominantly employed EMG signals and approached the problem as a classification task [32]–[37]. With the exception of [38] these studies have focused on categorizing muscle activity into discrete character classes rather than reconstructing the actual writing trajectories. Advancing [38], our work fundamentally shifts this paradigm by treating handwriting decoding as a regression problem, enabling continuous trajectory reconstruction that preserves the temporal and spatial dynamics of the original movement.

This regression-based approach offers several theoretical and practical advantages. First, it captures the rich temporal structure of handwritten trajectories that are lost in classification approaches. Second, it provides a more naturalistic interface for prosthetic control and HMIs that aligns with how the neuromuscular system inherently functions. Finally, it establishes a more challenging benchmark for evaluating the information content of myographic signals and the capability of neural architectures to extract this information.

Natural human movement unfolds through infinite gradations of force, speed and position, not through discrete, binary commands. When reaching for a delicate object, our muscles activate in complex, continuously varying patterns that precisely control pressure, trajectory and timing. Traditional prosthetic systems and HMIs that rely on discrete muscle signals create an artificial barrier between intention and action and force amputees to mentally translate their natural movement intentions into limited, predefined commands.

The continuous decoding paradigm demonstrated in this work addresses this fundamental mismatch by capturing the rich, analog nature of neuromuscular signals. By interpreting the subtleties of varying muscle activation patterns, our approach enables proportional control that mirrors natural movement. An amputee could potentially increase grip force gradually through intuitive, progressive muscle engagement rather than selecting between preset options.

Furthermore, this approach significantly reduces the cognitive burden on users, who would no longer need to consciously choose between limited movement categories. Instead, they could focus on the task itself and allow their natural muscle patterns to drive the prosthetic response. The resulting movements would appear more fluid, natural and potentially contributing to improved psychological acceptance and em-

bodiment of prosthetic devices.

The key limitation of this study is an absence of the direct comparison of OMG with EMG-based decoding, the current state-of-the-art in myographic recognitio system. However, current hardware constraints make it challenging to simultaneously collect. OMG armband requires tight skin contact to prevent external light interference through the bracelet's gaps. Integrating additional EMG electrodes at the same muscle sites would compromise the optical signal quality, making simultaneous acquisition currently infeasible with existing hardware. We are planning comparative studies that will include multiple sensing modalities for future work.

Notably, integration of OMG with existing myoelectric prosthetic systems should be relatively straightforward, as OMG sensors can directly replace conventional EMG sensors in current prosthetic designs without requiring major hardware modifications to the control architecture. However, full clinical deployment will still require addressing key engineering challenges including optimization of sensor placement within prosthetic sockets and development of user training protocols. Additionally, comparative evaluation against current clinical solutions will be essential to establish clinical efficacy and user acceptance. Future clinical development will require collaborative efforts with prosthetic manufacturers, rehabilitation specialists, and end-users to ensure that OMG-based systems meet the practical demands of daily prosthetic use while providing meaningful functional improvements over existing solutions.

If Wolpert is right about the brain's primary purpose being movement, then limb loss represents not just a physical challenge but a fundamental threat to how we experience being human. This is why prosthetic and HMI design must evolve beyond mere functionality to address both physical capability and psychological wellbeing.

## V. Conclusion

We demonstrated that OMG can be effectively leveraged to decode handwriting - an intricate motor activity through a regression-based approach, without requiring extensive preprocessing. Our TransScribe consistently outperformed the GRUScribe, which highlights the importance of modeling long-range dependencies in OMG signals for fine motor control decoding. Notably, the successful reconstruction of handwriting trajectories in amputee participants underscores the potential of our approach for developing prosthetic control paradigms that mirror natural human movement. Future work will focus on designing customized data augmentation techniques, reducing the dependence on large training datasets, and implementing low-latency, real-time decoding pipelines to advance toward clinically viable prosthetic applications.

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
