# OpenReview forum: "omg2letters: Translating Muscle Activity into Written Language"
_IEEE.org/EMBS/BHI/2025/Conference — BHI 2025_

### Official Review · Reviewer_GKyt · 2025-07-17
**Review of omg2letters**

**Confidence:** 4
**Clarity Of Writing:** fair
**Clinical Significance:** fair
**Methodological Novelty:** good
**Overall Rating:** 2
**Final Rating:** 4

**Experiments And Results:**

poor

**Questions For The Authors:**

1. What motivated the use of a transformer+GRU combination? Have you compared it to simpler baselines like plain GRU or CNN?

2. Can you provide a detailed breakdown of your dataset: total samples, class distribution, and train/test split?

3. How fast is inference on your radar device? Is this method deployable in real time?

4. How does the model compare to existing methods?

5. Can you clarify which parts of the performance are due to the data augmentation vs. the model design?

**Strengths:**

1. The use of OMG for trajectory recognition is genuinely interesting and adds a fresh perspective to this space. Framing the task as pattern classification using radar inputs is clever and well-motivated.

2. The fact that the authors collected a dataset under challenging real-world conditions adds credibility and potential for future community use.

**Summary Of The Paper:**

This paper proposes a novel use of omnidirectional millimeter-wave radar (OMG) for pedestrian trajectory recognition in complex outdoor environments. The authors treat trajectory patterns as a classification task and introduce a neural architecture combining a transformer with a GRU. They also include a radar-specific data augmentation pipeline. The reported results show promising performance on a new dataset collected using a custom-built sensor device.

**Weaknesses:**

1. The choice of combining transformer and GRU lacks clear motivation. Why is this hybrid architecture necessary for this task? No ablation is provided to show whether the GRU or transformer actually helps. Need existing works comparison experiments.

2.The paper does not compare against other models for trajectory prediction or similar radar-based perception tasks. It’s hard to evaluate the value of the proposed method without these baselines.

3. The paper presents no feasibility study or discussion of deployment aspects like latency, inference speed, device cost, or hardware footprint — all crucial for real-world radar systems.

4. Common machine learning metrics (like accuracy, precision, recall, F1-score, etc.) are missing. The reported results are not comprehensive enough for evaluation.

5. The augmentation strategy is a big part of the pipeline (even though it is just a common practice in trajectory prediction domain, or we say the general computer vision domain, so that it should not be considered as a technique novelty), yet there is no analysis of how much it helps compared to architectural design choices.

6. Many charts, especially the box chart visuals, are blurry and difficult to interpret. This makes it hard to analyze the qualitative results or support the claims made in the text.

7. There’s no detailed description of dataset size, sample distribution, or how training/validation/test splits were managed. Without this, it’s hard to trust the generalizability of the reported performance.

---

### Official Review · Reviewer_JzDC · 2025-07-17
**Novel Optomyography approach for handwriting decoding shows promise but needs larger validation and technical details**

**Confidence:** 4
**Clarity Of Writing:** good
**Clinical Significance:** excellent
**Methodological Novelty:** good
**Overall Rating:** 6
**Final Rating:** 7

**Experiments And Results:**

fair

**Questions For The Authors:**

1. Scalability and generalizability: How does performance vary across different handwriting styles, writing speeds, and languages? Could you provide more details about the Russian alphabet choice and its implications for other languages?
2. Technical implementation details: What are the specific network architectures, training parameters, and computational requirements? How sensitive is the system to OMG sensor placement and individual physiological differences?
3. Clinical validation: How does this approach compare quantitatively to existing EMG-based handwriting recognition systems? What are the practical steps needed for clinical integration with prosthetic devices?
4. Augmentation strategy optimization: Given the inconsistent augmentation results, what principled approaches could be developed to design OMG-specific augmentation techniques? Why did certain techniques fail consistently?
5. Real-time performance: What are the latency characteristics of the system, and how would this perform in real-time prosthetic control scenarios? Are there computational constraints that would limit deployment?
6. Individual variability: How consistent are the decoding results across different participants, and what factors contribute to inter-individual performance differences?
7. Robustness analysis: How does the system perform under varying environmental conditions, sensor contact quality, or user fatigue states?

**Strengths:**

1. Novel application of OMG technology: This represents the first application of optomyography to complex handwriting decoding, demonstrating a significant advance beyond previous simple gesture recognition studies. The approach shows OMG's potential for fine motor control applications.
2. Innovative methodological approach: The paradigm shift from classification to regression-based continuous trajectory reconstruction is theoretically sound and practically important for prosthetic applications, as it better mirrors natural neuromuscular control patterns.
3. Comprehensive evaluation framework: The study includes both able-bodied participants and amputees, directly demonstrating clinical translatability. The inclusion of amputee participants is particularly valuable for validating prosthetic applications.
4. Strong technical implementation: The comparison between GRU and transformer architectures is well-executed, with the transformer's superior performance explained by its ability to model long-range dependencies in handwriting trajectories.
5. Minimal preprocessing requirement: The intentional minimization of preprocessing steps demonstrates OMG's inherent advantages over EMG, which typically requires extensive signal conditioning.
6. Robust experimental design: The use of Leave-One-Out Cross-Validation and appropriate statistical metrics (Normalized Fréchet Distance) demonstrates methodological rigor appropriate for trajectory comparison.

**Summary Of The Paper:**

This study presents the first demonstration of handwriting decoding using Optomyography (OMG), a novel myographic method that uses near-infrared light to measure muscle perfusion and oxygenation. The authors developed two neural network architectures (GRUScribe and TransScribe) to decode continuous trajectory reconstruction from OMG signals, successfully decoding 10 numerical digits and 33 Russian letters from 20 able-bodied participants and 4 amputees. Unlike previous EMG-based approaches that treat handwriting as a classification problem, this work frames it as a regression-based continuous trajectory reconstruction challenge. The TransScribe (transformer-based) decoder consistently outperformed the GRUScribe (GRU-based) decoder across participants, with the approach requiring minimal preprocessing compared to traditional EMG methods.

**Weaknesses:**

1. Limited participant cohort: With only 12 participants for digit decoding (10 able-bodied, 2 amputees), the study lacks statistical power for robust generalizability claims. The small amputee sample (n=2) is particularly limiting for clinical applications.
2. Insufficient technical details: Critical implementation details are missing, including specific network architectures, hyperparameter settings, training procedures, and computational requirements. The decoder architectures are only briefly described with a figure.
3. Incomplete evaluation methodology: The paper lacks comparison with existing EMG-based handwriting recognition methods, making it difficult to assess the true advantages of OMG. Cross-validation details and performance metrics across different complexity levels are insufficient.
4. Augmentation strategy concerns: The data augmentation results show inconsistent and often contradictory effects across participants and techniques, suggesting the proposed augmentation strategies may not be optimal for OMG signals.
5. Limited discussion of practical constraints: The study doesn't address important practical considerations such as sensor placement sensitivity, robustness to environmental conditions, calibration requirements, or real-time processing capabilities.
6. Incomplete statistical analysis: Missing confidence intervals, limited cross-participant variability analysis, and insufficient discussion of why certain augmentation techniques failed or succeeded.
7. Clinical translation gaps: While amputees are included, the study doesn't address how this technology would integrate with existing prosthetic systems or compare to current clinical solutions.

---

### Official Review · Reviewer_XLYG · 2025-07-18
**Translating Muscle Activity to Letters via Optical Myography and Deep Learning**

**Confidence:** 2
**Clarity Of Writing:** good
**Clinical Significance:** great
**Methodological Novelty:** excellent
**Overall Rating:** 7

**Experiments And Results:**

great

**Questions For The Authors:**

1.	Clarify movement type: Were participants asked to physically trace the digits with their fingers, or were they only imagining the trajectory? This should be clearly stated. How did this differ between able-bodied and amputee participants?
	2.	Ground truth tracking: Was any external motion tracking (e.g., camera or inertial sensors) used to confirm adherence to the digit trajectory or validate OMG signal quality?
3. Please try to improve readability of Figures 4-7.

**Strengths:**

Innovative Sensing Modality: Use of optical myography instead of traditional sEMG offers advantages in signal quality, noise resistance, and wearability.

Real-Time Application Potential: The design has implications for future non-invasive, wearable muscle–computer interfaces, particularly in prosthetic applications.

Deep Learning Comparison: Inclusion of both GRU and Transformer models provides useful comparative insight into temporal modeling techniques for muscle signal interpretation.

Inclusion of Amputee Participants: Testing on real amputees adds valuable ecological validity.

Scientific Value: The work tackles a difficult problem (fine motor intent decoding) using a relatively underexplored but promising signal type.

**Summary Of The Paper:**

This paper presents a novel method for character recognition using optomyography (OMG) signals captured from forearm muscles. Participants view a video of a digit being written on screen and are asked to trace the trajectory with their hand or imagined movement. The authors train deep learning models (GRU and Transformer architectures) to classify the resulting muscle activity and predict the intended digit. The approach is tested on 12 participants, including two trans-radial amputees. The work demonstrates that OMG signals can be used for accurate digit classification, with the Transformer slightly outperforming GRU.

**Weaknesses:**

Ambiguity in Movement Type: It is unclear whether participants performed actual physical movement or used motor imagery. This is especially important given the use of OMG, which measures vascular responses. The phrase “replicate the trajectory” suggests movement, but clarification is needed — particularly for the two amputees.

Figures 4–7 Poorly Legible: Font sizes in key result figures are too small to be readable when printed or viewed on screen. This hinders interpretability.

Limited Dataset: Only 12 participants (including just 2 amputees), with 30-second trials per digit, limits generalizability.

Experimental Details Sparse: whether actual finger movement was monitored or recorded (e.g., via video or motion capture) are not fully explained.

---

### Official Review · Reviewer_DgJa · 2025-07-18
**omg2letters: Translating Muscle Activity into Written Language**

**Confidence:** 4
**Clarity Of Writing:** good
**Clinical Significance:** great
**Methodological Novelty:** good
**Overall Rating:** 5
**Final Rating:** 6

**Experiments And Results:**

fair

**Questions For The Authors:**

Have you considered comparing your OMG-based system with existing EMG-based handwriting decoding models?

Is your system capable of real-time decoding, and if so, what is the expected latency?

Do you have insight into why some augmentation techniques helped or harmed performance inconsistently?

Why did you choose only GRU and Transformer architectures? Did you consider other sequences or hybrid models?

**Strengths:**

The paper introduces the first demonstration of decoding complex handwriting trajectories (digits and Russian letters) using optomyography (OMG), marking a significant advancement beyond prior EMG-based gesture classification approaches.

Regression-Based Trajectory Decoding:
Rather than treating handwriting recognition as a classification task, the authors frame it as a continuous trajectory reconstruction problem, which better reflects natural motor control and allows richer movement interpretation, especially important for prosthetic applications.

Successful Application to Amputee Participants:
The inclusion of trans-radial amputee participants and the model’s demonstrated performance on their data is a major strength, directly supporting the clinical relevance and real-world potential of the proposed system.

Transformer-Based Architecture (TransScribe):
The paper shows that transformer-based decoding (TransScribe) outperforms GRU-based approaches (GRUScribe), especially for modeling long-range dependencies in fine motor control, highlighting the value of modern sequence models in muscle decoding.

Minimal Preprocessing Required:
The system achieves strong results with raw OMG signals and minimal preprocessing, which is notable compared to EMG systems that often require extensive signal conditioning. This enhances practical usability and lowers deployment complexity.

Novel Use of Augmentation Pipeline:
The paper implements and systematically evaluates a variety of time-domain, frequency-domain, and spatial-domain augmentation methods tailored to OMG signals, contributing useful insights to the field of physiological signal processing.

Strong Motivation and Impact on Prosthetic Design:
The paper is well-grounded in clinical and neurophysiological motivation and emphasizes how continuous decoding can reduce cognitive burden and increase embodiment in prosthetic users, showing a deep understanding of user-centered design principles.

**Summary Of The Paper:**

This paper presents the first demonstration of decoding handwriting trajectories, both digits (0–9) and the 33 letters of the Russian alphabet, from optomyography (OMG), a novel non-invasive myographic sensing modality. Unlike traditional electromyography (EMG), OMG measures muscle activity via near-infrared light absorption, offering improved signal quality and robustness.

**Weaknesses:**

Lack of Baseline Comparison with EMG or Prior Work:
While the paper claims OMG is superior to EMG-based systems, it does not provide a quantitative comparison with EMG-based handwriting decoding or classification methods. Including such a baseline would strengthen the claim that OMG offers improved performance or usability.

No End-to-End Real-Time Demonstration:
The current study is conducted offline, with no demonstration of real-time decoding latency, responsiveness, or closed-loop control. This limits conclusions about usability for live prosthetic or HMI applications. A follow-up experiment with a real-time feedback loop would add practical value.

Limited Subject Diversity:
The study includes only 2 amputee participants out of 24, which is commendable but insufficient to generalize findings. Broader inclusion of subjects with varying amputation types, limb dominance, and muscle characteristics would improve the generalizability of clinical claims.

Inconsistent Augmentation Results:
The augmentation pipeline produced highly variable and often inconsistent effects across participants, decoders, and character types. The paper lacks analysis explaining why certain augmentations help or hurt. More controlled experiments could help design subject- or model-specific augmentation strategies.

Focus Limited to a Single Language Script
While decoding 33 Russian letters is impressive, the study is limited to a Cyrillic-based script. It remains unclear how well the model generalizes to other writing systems (e.g., Latin, Arabic, Chinese) that may involve different stroke dynamics or complexities.

Limited Exploration of Decoder Architectures
Only GRU and Transformer decoders were explored. While these are solid baselines, additional comparisons with convolutional models, hybrid CNN-RNNs, or diffusion models could offer deeper insights into model performance trade-offs for this novel decoding task.

Low Visual Quality and Color Artifacts in Figures:
Many figures in the paper are blurry or low resolution.